# Sexual dimorphism of the human fetal pelvis exists at the onset of primary ossification
Toru Kanahashi [1] ✉, Jun Matsubayashi [2], Hirohiko Imai[3], Shigehito Yamada [1,4], Hiroki Otani [5] & Tetsuya Takakuwa [1]

Human adolescent and adult skeletons exhibit sexual dimorphism in the pelvis. However, the degree of sexual dimorphism of the human pelvis during prenatal development remains unclear. Here, we performed high-resolution magnetic resonance imaging-assisted pelvimetry on 72 human fetuses (males [M]: females [F], 34:38; 21 sites) with crown-rump lengths (CRL) of 50–225 mm (the onset of primary ossification). We used multiple regression analysis to examine sexual dimorphism with CRL as a covariate. Females exhibit significantly smaller pelvic inlet anteroposterior diameters (least squares mean, [F] 8.4 mm vs. [M] 8.8 mm, $P = 0.036$), larger subpubic angle ([F] 68.1° vs. [M] 64.0°, $P = 0.034$), and larger distance between the ischial spines relative to the transverse diameters of the greater pelvis than males. Furthermore, the sacral measurements indicate significant sex-CRL interactions. Our study suggests that sexual dimorphism of the human fetal pelvis is already apparent at the onset of primary ossification.

Human adolescent and adult skeletons exhibit sexual dimorphism in the teeth, cranium, pelvis, and mandible[1]. In particular, the adult pelvis displays a considerable degree of sexual dimorphism, and the hip bones (ilium, pubis, and ischium) are frequently used in the sex determination of forensic specimens[2]. Several studies on pelvic sexual dimorphism have reported sex differences in the pelvic inlet and outlet, subpubic angle, greater sciatic notch, pubic length, bi-iliac breadth, and sacral width[3–16].

However, research on pelvic sexual dimorphism during fetal and juvenile development is limited and has conflicting results. It is unclear whether significant sexual dimorphism is present prenatally; nevertheless, a consensus exists that the differences are insubstantial for reliable determination of sex from the hipbones[17]. Hromada[18] noted no sex differences in the ratios of the sciatic notch, ilium, and pelvic inlet and outlet from the second to the 7th month of pregnancy (10–30 weeks gestation); however, he found a sex difference from the 7th month (30 weeks gestation) until birth. Boucher[19] reported sexual dimorphism in the greater sciatic notch and subpubic angle in fetus specimens 15–45 weeks gestation. Weave[20] found no significant differences in the greater sciatic notch morphology between late fetal periods (28–36 weeks gestation) and children, using the same measurements and proposed alternatives.

Holcomb et al.[21] reported sexual dimorphism in the greater sciatic notch in fetus specimens >16 weeks gestation. However, Mokrane et al.[22] conducted a comprehensive three-dimensional (3D) morphometric analysis based on reconstructed computed tomography images of the fetal ilium ranging from 21 to 40 weeks gestation and reported no significant sex differences. Haque[23] reported sex differences in the subpubic angle among fetus specimens ranging from 14 to 22 weeks gestation with a crown-rump length (CRL) of 120–210 mm[24].

In the ilial and sacral bones, primary ossification begins 9 weeks post-fertilization (approximate CRL of 50 mm)[1,24,25]. Most previous analyses have focused on the late fetal period when the primary ossification of the pelvic bones has progressed to some extent[17]. However, no prior study has investigated pelvic sexual dimorphism at the onset of primary ossification. Moreover, numerous studies have focused on the ilium because it is easily observable and is a location where sex differences are observed in adults[21]. However, analyzing sexual dimorphism in the morphology of individual cartilages, apart from the ilium, is inadequate when evaluating the fetal pelvis. The pelvis is a distinctive site where multiple bones, including the iliac, sciatic, pubic, and sacral/caudal bones, fuse to form a complex structure. Therefore, it is crucial to investigate

[1]Human Health Science, Graduate School of Medicine, Kyoto University, Kyoto, Japan. [2]Center for Clinical Research and Advanced Medicine, Shiga University of Medical Science, Shiga, Japan. [3]Department of Informatics, Graduate School of Informatics, Kyoto University, Kyoto, Japan. [4]Congenital Anomaly Research Center, Graduate School of Medicine, Kyoto University, Kyoto, Japan. [5]Department of Developmental Biology, Faculty of Medicine, Shimane University, Shimane, Japan. ✉e-mail: kanahashi.toru.7e@kyoto-u.ac.jp

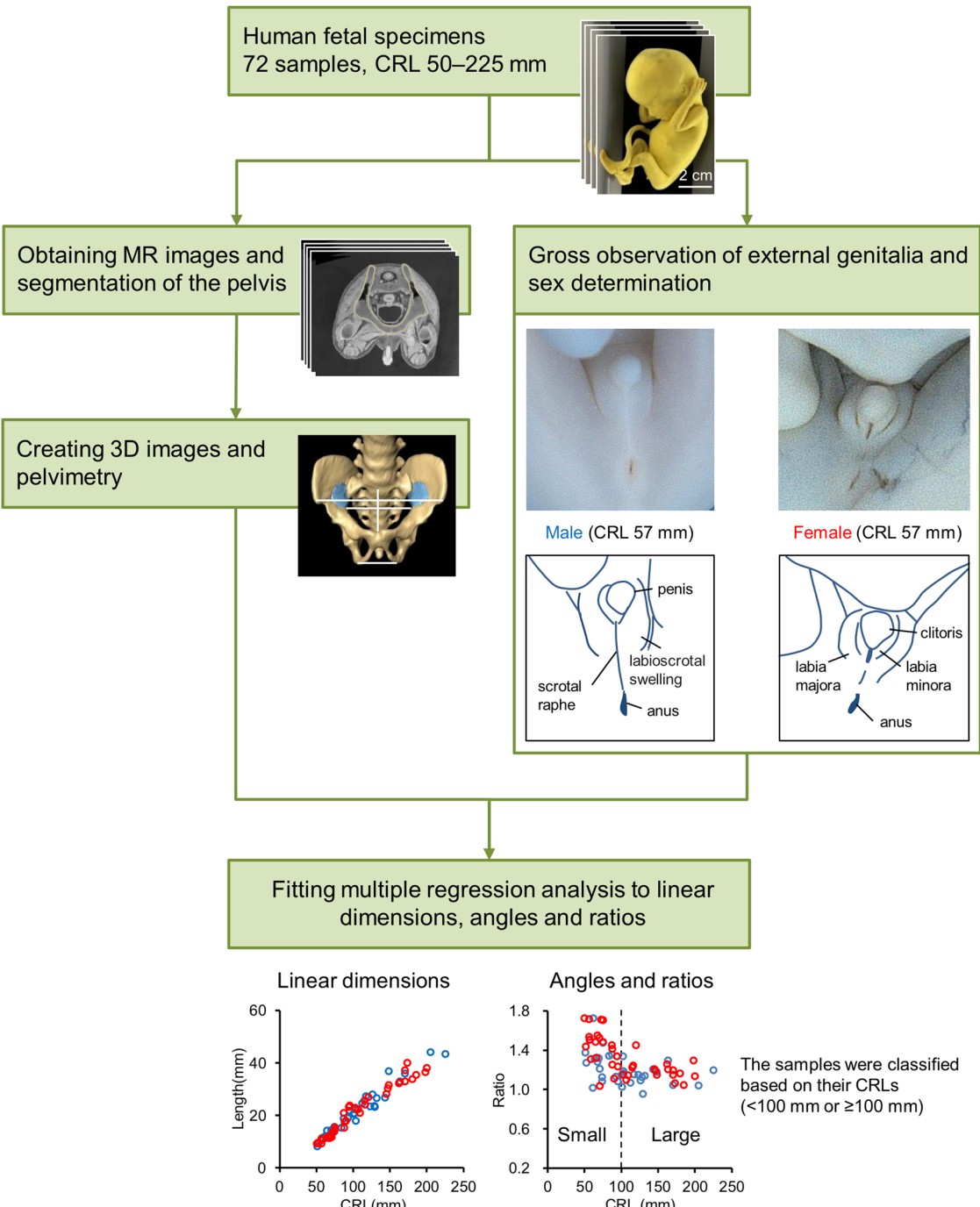

**Fig. 1 | Workflow for sexual dimorphism analysis of the pelvis of human fetus specimens.** Pelvic segmentation was performed on magnetic resonance (MR) images of 72 human fetuses stored at the congenital anomaly research center of Kyoto University and Shimane University, followed by three-dimensional (3D) reconstruction and pelvimetry. The sex was determined through observation of the external genitalia. Pelvic sexual dimorphism was assessed using multiple regression analysis. CRL crown-rump length.

sexual dimorphism in the proportion and morphology of the entire pelvis.

Acquiring human fetus specimens for research poses challenges due to ethical and other concerns[26], making them highly valuable resources. Given their importance, it is essential to analyze the human feal specimens using non-destructive methods. High-resolution magnetic resonance imaging (MRI), a non-destructive analysis technique enabling accurate 3D reconstruction, is being used to analyze organ morphogenesis in human fetal specimens[25,27].

In this study, we aimed to investigate pelvic sexual dimorphism using T1-weighted MR images in human fetus specimens with a CRL from 50 mm, which corresponds to the onset of primary ossification, to 225 mm and 3D reconstruction of T1-weighted MR images for pelvimetric measurements. The sex of human fetuses can be accurately determined from their external genitalia after a CRL of 50 mm[28]. Our results indicated significant sexual dimorphism of the human fetal pelvis at multiple sites during the early fetal period. Females had significantly smaller anteroposterior diameters of the pelvic inlet, larger subpubic angle, and larger distances

**Table 1 | Description of the linear dimensions and angles in the pelvis**

| ID | Description | Ratio |
|---|---|---|
| **Linear dimensions of the greater pelvis** | | |
| #1 | Length between the furthest points of the two iliac crests (intercristal distance) | |
| #2 | Length between the bilateral ASIS (interspinous distance) | |
| #3 | Length between the bilateral acetabula | #3/#1, #3/#2 |
| #4 | Length between the ASIS and PSIS (lateral conjugate) | |
| #5 | Length between the center of the acetabulum and the superomedial border of the pubic symphysis (superior ramus of pubis length) | |
| #6 | Length between the superomedial border and inferomedial border of the pubic symphysis (pubic symphysis length) | |
| #7 | Length between the most superior point on the iliac crest and the most inferior point on the ischial tuberosity (os coxa length) | |
| #8 | Length between the center of the acetabulum and the most superior point on the iliac crest (iliac blade height) | #8/#7 |
| #9 | Length between the posterior inferior iliac spine and ischial spine (sciatic notch width) | |
| #10 | Length between the center of the acetabulum and the most inferior point on the ischial tuberosity | #10/#7, #10/#8 |
| **Linear dimensions of the lesser pelvis** | | |
| #11 | Maximum transverse diameter of the pelvic inlet | #11/#1, #11/#2, #11/#12 |
| #12 | Antero-posterior diameters of the pelvic inlet | |
| #13 | Length between the bilateral ischial spines | #13/#1, #13/#2, #13/#3, #13/#11, #13/#14 |
| #14 | Length between bilateral ischial tuberosities (intertuberous distance) | #14/#1, #14/#2, #14/#3, #14/#11 |
| **Linear dimensions of the sacrum** | | |
| #15 | Length between the sacral promontory and the midpoint of the third sacral vertebra (superior sacrum) | |
| #16 | Length between the midpoint of the third sacral vertebra and the inferior border of the fifth sacral vertebra (inferior sacrum) | |
| #17 | Transverse diameter of the first sacral vertebra | #17/(#15 + #16) |
| #18 | Transverse diameter of the third sacral vertebra | #18/(#15 + #16) |
| #19 | Transverse diameter of the fifth sacral vertebra | #19/(#15 + #16) |
| **Angle** | | |
| #20 | Iliac crest angle | |
| #21 | Subpubic angle | |

*ASIS* anterior superior iliac spine, *PSIS* posterior superior iliac spine.

**Fig. 2 | Linear dimensions, angles, and ratios calculated from 3D reconstructed images in pelvimetry.** The solid lines indicate the distance between the two observable points (colored in red), and the dotted line indicates the distance between the opposite sides, which is hidden. **a** Linear dimensions of the greater pelvis (a part) in ventral view. **b** Linear dimensions of the greater pelvis (a part) in the right lateral view. **c** Linear dimensions of the lesser pelvis. **d** Linear dimensions of the sacrum. **e** Iliac crest angle as depicted in the cranial view. **f** Subpubic angle as depicted in the ventral view. ASIS anterior superior iliac spine, IS ischial spine, IT ischial tuberosity, PIIS posterior inferior iliac spine, PSIS posterior superior iliac spine.

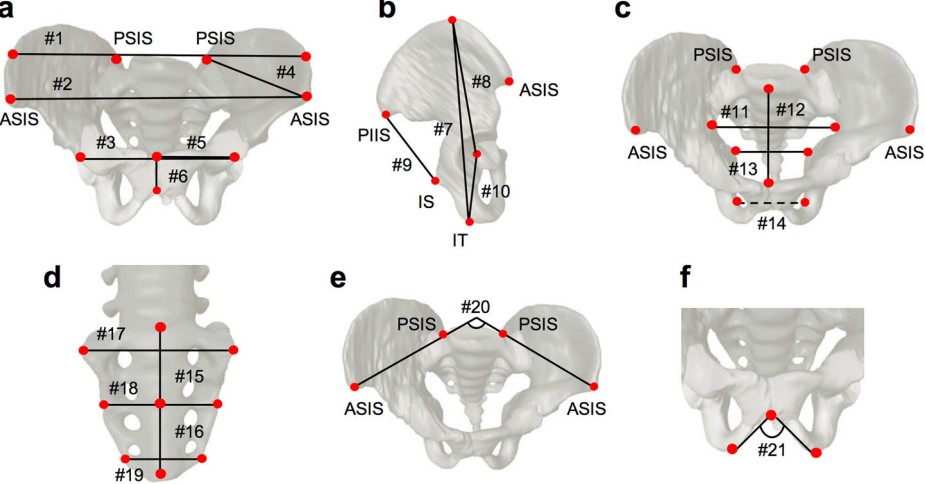

between the ischial spines relative to the two transverse diameters of the greater pelvis than males. Our study provides critical evidence-based insights into the emergence of sexual dimorphism in the human fetal pelvis earlier than previously reported, using simple pelvimetry and statistical modeling methods.

## Results
### Workflow overview
The workflow for analyzing sexual dimorphism of the human fetal pelvis is presented in Fig. 1. Seventy-two human fetus specimens were categorized based on external genitalia morphology as males ($n$ = 34; CRL:

**Fig. 3 | Sexual dimorphism in the linear pelvic dimensions. a** Least squares mean and 95% confidence intervals for the sex differences in the linear dimensions of the greater and lesser pelvis. Positive least squares mean values indicate that females exhibit larger measurements. The asterisks and black circles indicate significant differences (*P* < 0.05). The dataset comprised 72 human fetus specimens. **b** Scatter plot of the anteroposterior diameters of the pelvic inlets (#12). The male and female regression lines are represented in blue and red, respectively. **c** Cranial view of the reconstructed pelvic inlet in males (CRL, 64 mm) and females (CRL, 65 mm). The light blue areas indicate the areas of primary ossification. Scale bar, 2 mm. CRL crown-rump length.

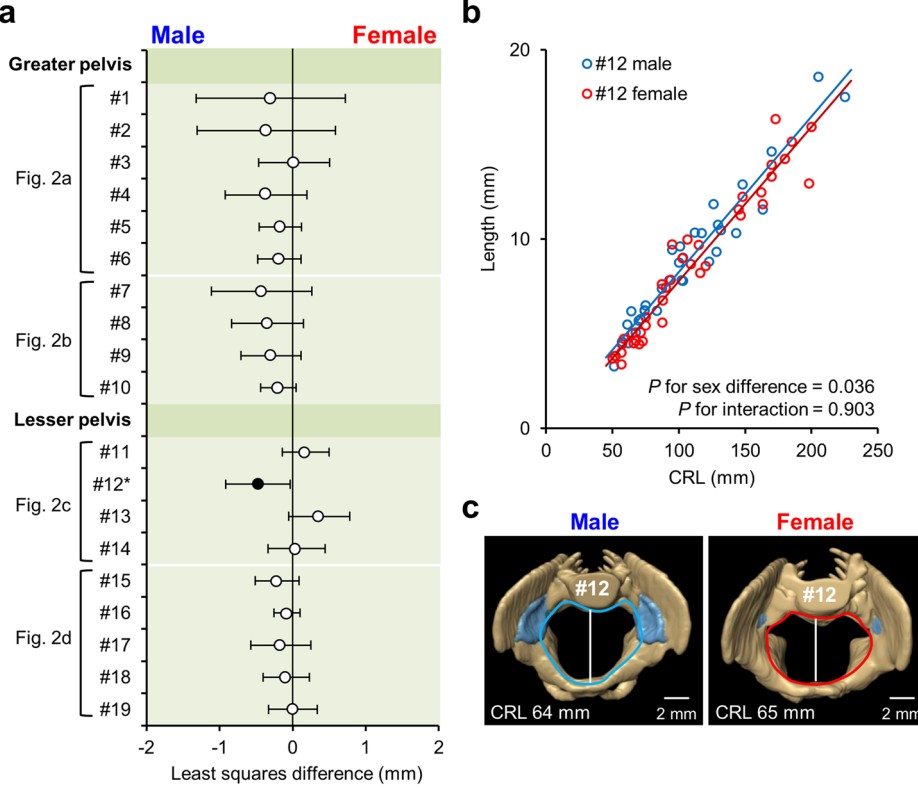

**Fig. 4 | Sexual dimorphism in pelvic angles. a** Least squares mean and 95% confidence intervals for sex differences in the iliac crest and subpubic angle. Positive least squares mean values indicate that females exhibit larger measurements. The asterisks and black circles indicate significant differences (*P* < 0.05). The dataset comprised 72 human fetus specimens. **b** Scatterplot of the subpubic angle (#21). Blue and red squares indicate the estimated means for males and females in each CRL subgroup, respectively. **c** Ventral view of the subpubic angle in males (CRL, 71 mm) and females (CRL, 71 mm). Scale bar, 1 mm. CRL crown-rump length.

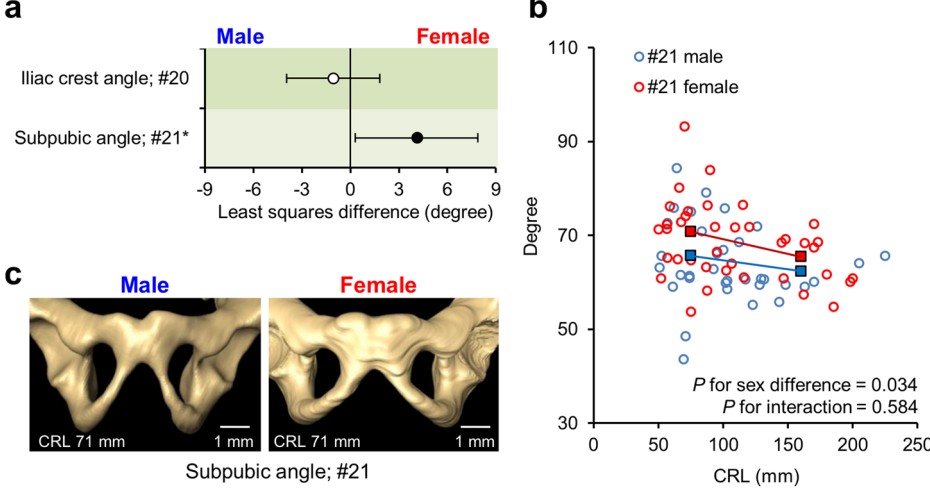

51–225 mm) and females (*n* = 38; CRL: 50–200 mm). The T1-weighted MR images were acquired using either one of the two 7-T MRI systems or the 3-T MRI system. The 3D images of the pelvis were manually reconstructed from these scans (Supplementary Movies 1–4). The linear dimensions of 19 specific sites and two angles were measured (Fig. 2 and Table 1 present a detailed illustration and description of the measurement sites, respectively), and 20 ratios based on the linear dimensions were calculated to assess the overall pelvic morphology. Finally, a multiple regression analysis was performed to examine the sex differences in pelvic morphology with CRL as a covariate.

## Pelvimetry and ratios
Table 2 summarizes the linear pelvic dimensions and angles. Strong positive correlations were observed between the linear dimensions and CRL in males and females (0.90–0.98). The scatter plots are presented in Supplementary

Figs. 1–4. These findings supported the use of a regression approach with CRL as a continuous covariate. Conversely, moderate or weak correlations, ranging from −0.38 to 0.63, were observed between the pelvic angles and CRL (scatter plots are presented in Supplementary Fig. 5).

Table 3 summarizes the pelvimetry ratios. Moderate or weak correlations, ranging from −0.65 to 0.54, were observed between the pelvic angles and CRL. The scatter plots (Supplementary Figs. 6–11) demonstrate a considerable difference in the association between CRL and ratios in smaller specimens compared with larger specimens.

## Sexual dimorphism in linear pelvic dimensions
We investigated sex differences in linear pelvic dimensions using multiple regression analysis. This analysis incorporated sex and CRL (as a continuous variable) as independent factors along with their interactions (Fig. 3) (detailed estimates are provided in Supplementary Table 1). The

## Table 2 | Summary of pelvimetry

| Measurement | Male (n = 34) | | Female (n = 38) | |
|---|---|---|---|---|
| | Mean (SD) | Correlation coefficient[a] | Mean (SD) | Correlation coefficient[a] |
| **Linear dimensions of the greater pelvis (mm)** | | | | |
| #1 | 21.5 (9.2) | 0.97 | 22.1 (9.8) | 0.98 |
| #2 | 20.6 (8.9) | 0.97 | 21.1 (9.6) | 0.98 |
| #3 | 10.0 (4.4) | 0.97 | 10.4 (4.4) | 0.98 |
| #4 | 12.8 (5.4) | 0.98 | 12.9 (5.6) | 0.98 |
| #5 | 6.4 (2.7) | 0.98 | 6.5 (2.8) | 0.97 |
| #6 | 3.1 (1.4) | 0.90 | 3.0 (1.4) | 0.90 |
| #7 | 17.7 (7.3) | 0.98 | 17.9 (8.0) | 0.98 |
| #8 | 12.0 (4.9) | 0.98 | 12.1 (5.3) | 0.98 |
| #9 | 8.4 (3.2) | 0.96 | 8.4 (3.3) | 0.97 |
| #10 | 5.9 (2.6) | 0.98 | 6.0 (2.8) | 0.98 |
| **Linear dimensions of the lesser pelvis (mm)** | | | | |
| #11 | 10.2 (3.7) | 0.97 | 10.7 (4.0) | 0.97 |
| #12 | 8.7 (3.6) | 0.97 | 8.5 (3.9) | 0.97 |
| #13 | 6.7 (2.9) | 0.95 | 7.3 (3.3) | 0.96 |
| #14 | 6.5 (2.8) | 0.97 | 6.8 (2.8) | 0.95 |
| **Linear dimensions of the sacrum (mm)** | | | | |
| #15 | 5.9 (2.2) | 0.97 | 5.9 (2.6) | 0.96 |
| #16 | 4.0 (1.6) | 0.98 | 4.1 (1.8) | 0.97 |
| #17 | 10.2 (4.0) | 0.98 | 10.4 (4.2) | 0.98 |
| #18 | 8.0 (2.9) | 0.98 | 8.2 (3.3) | 0.97 |
| #19 | 5.7 (2.0) | 0.95 | 5.9 (2.6) | 0.96 |
| **Angles (degree)** | | | | |
| #20 | 62.9 (7.2) | 0.63 | 61.4 (7.7) | 0.63 |
| #21 | 63.9 (8.3) | -0.17 | 68.2 (8.1) | -0.38 |

*SD* standard deviation.

[a]Pearson correlation coefficient of crown-rump length and pelvimetric values.

anteroposterior diameter of the pelvic inlet (#12) was significantly shorter in females than in males (least squares difference [95% confidence interval], −0.5 mm [−0.9 mm, 0.0 mm], P = 0.036) (Fig. 3, Supplementary Table 1, Supplementary Videos 1, 2).

### Sexual dimorphism in pelvic angles

Sex differences in the pelvic angles were examined using multiple regression analysis, which included sex, subgroups based on CRL (specimens with CRL < 100 mm and ≥100 mm), and their interaction as independent variables (Fig. 4 and Supplementary Table 2). The subpubic angle (#21) was significantly larger in females than in males (least squares differences [95% confidence interval], 4.1° [0.3°, 7.9°], P = 0.034). Furthermore, this was significantly smaller in the subgroup with CRL ≥ 100 mm (−4.3° [−8.1°, −0.5°], P = 0.026) (Fig. 4, Supplementary Table 2, Supplementary Videos 3, 4). To confirm the reliability of the sex difference identified in the subpubic angle (#21), we employed two different types of multiple regression analysis. Both analyses incorporated the CRL as a continuous variable, with one model incorporating a curve by introducing a squared CRL term. These additional regression analyses yielded consistent results (Supplementary Tables 4, 5).

### Sexual dimorphism in pelvic ratios

Sex differences in pelvic ratios were examined using the same regression model as used for pelvic angles (Fig. 5 and Supplementary Table 3). The pelvic inlet ratio (#11/#12, ratio of transverse diameter to the anteroposterior diameter) and a part of the ratio of transverse diameters related

to the ischial spine (#13/#1, ratio of the distance between the ischial spines to the intercristal diameter; #13/#2, ratio of the distance between the ischial spines to the interspinous diameter) were significantly larger in females than in males (least squares differences [95% confidence interval], 0.10 [0.03, 0.17], P = 0.006; 0.02 [0.00, 0.04], P = 0.019; and 0.03 [0.01, 0.05], P = 0.009, respectively) (Fig. 5, Supplementary Table 3, Supplementary Videos 1, 2). The additional regression analyses yielded consistent results (Supplementary Tables 4, 5). All the aforementioned ratios were significantly smaller in the subgroup with CRL≥100 mm (−0.20 [−0.27, −0.13], P < 0.001; −0.02 [−0.04, 0.00], P = 0.047; and −0.02 [−0.04, −0.01], P = 0.014, respectively) (Supplementary Table 3).

### Interaction between sex and CRL in the sacrum

Significant interactions between sex and CRL were only found in the sacrum (Fig. 6). The interaction was significant in the transverse diameter of the fifth sacral vertebra (#19) (P = 0.042) (Fig. 6, Supplementary Table 1), indicating that the CRL coefficient for #19 was higher (i.e., the slope of the regression line was steeper) in females than in males. The interaction between sex and subgroups based on CRL was significant in the #17/(#15 + #16) ratio related to the sacrum (P = 0.027) (Fig. 6, Supplementary Table 3), indicating that the ratio for females was larger than that for males in the subgroup with CRL < 100 mm, while no sexual differences were found in the subgroup with CRL ≥ 100 mm.

### Discussion

In this study, we used pelvimetry and multiple regression analysis to demonstrate the significant differences in the anteroposterior diameter of the pelvic inlet, subpubic angle, and the ratio of the distance between the ischial spines and the transverse diameter of the greater pelvis between male and female fetus specimens with CRLs of 50–225 mm. In addition, significant interactions between sex and CRL were found in the sacral measurements (transverse diameter of the fifth sacrum and the ratio of the transverse diameter of the first sacrum to the sacrum's longitudinal length), suggesting sexual dimorphism during the prenatal development of the sacrum. The sexual dimorphism of the fetal pelvis demonstrated in this study was comparable to that observed in the adult pelvis, except for the anteroposterior diameter of the pelvic inlet.

The transverse to anteroposterior diameter ratio at the pelvic inlet was significantly higher in females than in males, indicating that the relative transverse diameters were more significant in females than in males, which was similar to sexual dimorphism of the adult pelvic inlet[29–31] The common textbook description of the shape of the male and female pelvic inlet is an android (triangle) and gynoid (transverse oval), respectively[32]. Observations of the 3D images of the male and female pelvis in this study confirmed a similar shape during the fetal period. However, the discrepancies in the anteroposterior and transverse diameters of the pelvic inlet between the male and female fetuses differed from those in adults. In adults, the anteroposterior and transverse diameters of the inlet are significantly larger in females than in males[30,31].

In the present study, the anteroposterior diameters of the inlet were significantly larger in males than in females; however, the reason for this significance is unclear. No significant differences were observed in the transverse diameters of the inlet between the sexes. Thus, significant sex differences in the anteroposterior diameters may have had a major influence on the significant differences in the ratio at the inlet. The dilatation of the pelvic canal in adult females may be developmentally induced by the volume and location of the reproductive pelvic organs (vagina, ovaries, and uterus)[33]. Therefore, it is necessary to investigate whether the location and size of the reproductive organs, rectum, bladder, and other pelvic organs affect sex differences in anteroposterior diameters in the fetal period.

The subpubic angle (#21) exhibits sexual dimorphism in adults[4,5,29]. Previous studies that examined sexual dimorphism of the subpubic angle in human fetuses from approximately 14 weeks gestation to birth reported significant sexual dimorphism from 14 to 22 weeks gestation (CRL: 120–210 mm), with no significant difference observed after

**Table 3 | Summary of pelvimetry ratios**

| Measurement | Male (*n* = 34) | | | Female (*n* = 38) | | |
|---|---|---|---|---|---|---|
| | Median (Q1, Q3) | Mean (SD) | Correlation coefficient[a] | Median (Q1, Q3) | Mean (SD) | Correlation coefficient[a] |
| **Ratio of the pelvic inlet** | | | | | | |
| $\frac{\#11}{\#12}$ | 1.17 (1.09,1.32) | 1.21 (0.16) | −0.43 | 1.26 (1.15, 1.49) | 1.33 (0.20) | −0.65 |
| **Ratios of the transverse diameters of the pelvis** | | | | | | |
| $\frac{\#3}{\#1}$ | 0.46 (0.44, 0.48) | 0.46 (0.03) | −0.04 | 0.48 (0.45, 0.50) | 0.48 (0.03) | −0.30 |
| $\frac{\#11}{\#1}$ | 0.47 (0.45, 0.52) | 0.49 (0.06) | −0.57 | 0.49 (0.45, 0.56) | 0.51 (0.07) | −0.66 |
| $\frac{\#13}{\#1}$ | 0.32 (0.29, 0.33) | 0.31 (0.04) | −0.10 | 0.35 (0.30, 0.37) | 0.34 (0.04) | −0.17 |
| $\frac{\#14}{\#1}$ | 0.30 (0.28, 0.33) | 0.30 (0.03) | −0.07 | 0.32 (0.29, 0.34) | 0.31 (0.03) | −0.44 |
| $\frac{\#3}{\#2}$ | 0.48 (0.46, 0.50) | 0.49 (0.04) | −0.09 | 0.51 (0.47, 0.52) | 0.50 (0.04) | −0.43 |
| $\frac{\#11}{\#2}$ | 0.49 (0.48, 0.55) | 0.51 (0.06) | −0.60 | 0.51 (0.48, 0.60) | 0.54 (0.08) | −0.71 |
| $\frac{\#13}{\#2}$ | 0.33 (0.30, 0.34) | 0.33 (0.04) | −0.12 | 0.36 (0.31, 0.40) | 0.36 (0.05) | −0.27 |
| $\frac{\#14}{\#2}$ | 0.31 (0.29, 0.34) | 0.32 (0.03) | −0.11 | 0.34 (0.31, 0.36) | 0.33 (0.04) | −0.51 |
| $\frac{\#13}{\#3}$ | 0.68 (0.64, 0.71) | 0.68 (0.07) | −0.07 | 0.70 (0.66, 0.75) | 0.71 (0.07) | 0.02 |
| $\frac{\#14}{\#3}$ | 0.64 (0.62, 0.69) | 0.66 (0.05) | −0.03 | 0.66 (0.62, 0.70) | 0.66 (0.06) | −0.30 |
| $\frac{\#13}{\#11}$ | 0.66 (0.57, 0.70) | 0.64 (0.08) | 0.44 | 0.67 (0.60, 0.73) | 0.67 (0.09) | 0.47 |
| $\frac{\#14}{\#11}$ | 0.63 (0.60, 0.67) | 0.62 (0.07) | 0.54 | 0.63 (0.56, 0.69) | 0.63 (0.08) | 0.26 |
| $\frac{\#13}{\#14}$ | 1.03 (0.97, 1.09) | 1.04 (0.11) | −0.04 | 1.07 (1.00, 1.15) | 1.08 (0.11) | 0.27 |
| **Ratios of the cranial-to-caudal diameter of the pelvis** | | | | | | |
| $\frac{\#8}{\#7}$ | 0.68 (0.67, 0.69) | 0.68 (0.01) | 0.18 | 0.67 (0.66, 0.69) | 0.68 (0.02) | −0.12 |
| $\frac{\#10}{\#7}$ | 0.33 (0.33, 0.34) | 0.33 (0.01) | 0.35 | 0.33 (0.33, 0.34) | 0.33 (0.01) | 0.30 |
| $\frac{\#10}{\#8}$ | 0.49 (0.47, 0.51) | 0.49 (0.02) | 0.22 | 0.49 (0.48, 0.51) | 0.49 (0.03) | 0.26 |
| **Aspect ratios length-to-width of the sacrum** | | | | | | |
| $\frac{\#17}{\#15+\#16}$ | 1.02 (0.98, 1.07) | 1.03 (0.07) | 0.08 | 1.05 (1.00, 1.10) | 1.06 (0.08) | −0.50 |
| $\frac{\#18}{\#15+\#16}$ | 0.81 (0.78, 0.84) | 0.81 (0.06) | −0.21 | 0.83 (0.81, 0.87) | 0.84 (0.07) | −0.57 |
| $\frac{\#19}{\#15+\#16}$ | 0.58 (0.53, 0.63) | 0.59 (0.07) | −0.32 | 0.61 (0.56, 0.64) | 0.60 (0.06) | −0.23 |

*Q1* first quartile, *Q3* third quartile, *SD* standard deviation.
[a]Pearson's correlation coefficient of crown-rump length and pelvimetry ratio.

23 weeks gestation[23]. The present study provides evidence that sexual dimorphism in the subpubic angle emerges earlier than previously reported. Washburn[11] noted that adult females have a wider subpubic angle than adult males because of pubic elongation. The present study found a significantly wider subpubic angle in females, whereas no sex differences in pubis length were observed. These findings suggest that the sex differences in the subpubic angle during the fetal period may not be due to pubic elongation. This study found that the transverse diameter of the middle pelvic cavity, where the ischial spine is located, relative to the transverse diameters of the greater pelvis (#13/#1, #13/#2), was significantly higher in females than in males. This may be related to the larger subpubic angle observed in females than in males.

The distance between the bilateral ischial spines and that between the ischial tuberosities are both significantly greater in adult females than in adult males[34]. In the present study, no sex differences were observed in the distance between the ischial spines. However, a significant sex difference was observed in the relative distance between the ischial spines and the transverse diameter of the greater pelvis, a trend akin to that observed in adults.

Boucher[19] reported sex differences in the width of greater sciatic notch in fetus specimens. However, the shape of the greater sciatic notch is a smooth continuum[21,22]; thus, determining a measurement reference point using Boucher's method and obtaining accurate measurements is challenging. Precise analysis of the sexual dimorphism of the greater sciatic notch requires the acquisition of objective landmark coordinates that can capture the entire shape of the greater sciatic notch[20–22]. In the present study, the measurement sites were selected based on distinct anatomical landmarks. The distance between the inferior posterior iliac and ischial spine landmarks

was measured to assess the width of the greater sciatic notch (#9); however, no sex differences were found.

No significant differences were found in the linear dimension of the transverse diameter of the fifth sacral vertebra (#19) or in the ratio of the transverse diameter of the first sacral vertebra to the longitudinal diameter of the sacrum (#17/[#15 + #16]) between the sexes. However, significant interactions were observed between sex and CRL, indicating that the coefficients of CRL differ significantly between males and females. These significant interactions suggest differences in changes in sacral morphology between sexes during the prenatal period. Tague[35] reported that the width of adult female sacral vertebrae exceeded those of adult male sacral vertebrae. The ratios of the transverse diameters of the first, third, and fifth sacral vertebrae to the longitudinal diameter of the sacrum were higher in females than in males in the present study, albeit not statistically significant. In addition, compared with males, females had a significantly larger coefficient of the CRL in the linear dimension of the transverse diameter of the fifth sacral vertebra (#19). The sacral ratio index used in the present study may aid in understanding sacral sexual dimorphism during late fetal and postnatal development.

Animal model developmental studies have revealed that pelvic sexual dimorphism is primarily determined by the spatial distribution of estrogen, androgen, and relaxin hormone receptors and hormone-induced bone remodeling[36,37]. Sertoli and Leydig cells differentiate and secrete steroid hormones from the 8th and 9th weeks of gestation in humans[38,39]. In addition, the masculinization programming window in humans lasts from the 8th to 14th weeks of gestation[39]. Since this study was conducted on fetuses over a gestational age of 9 weeks, sex hormones may have influenced dimorphic growth and pelvic remodeling. Steroid hormones are involved in

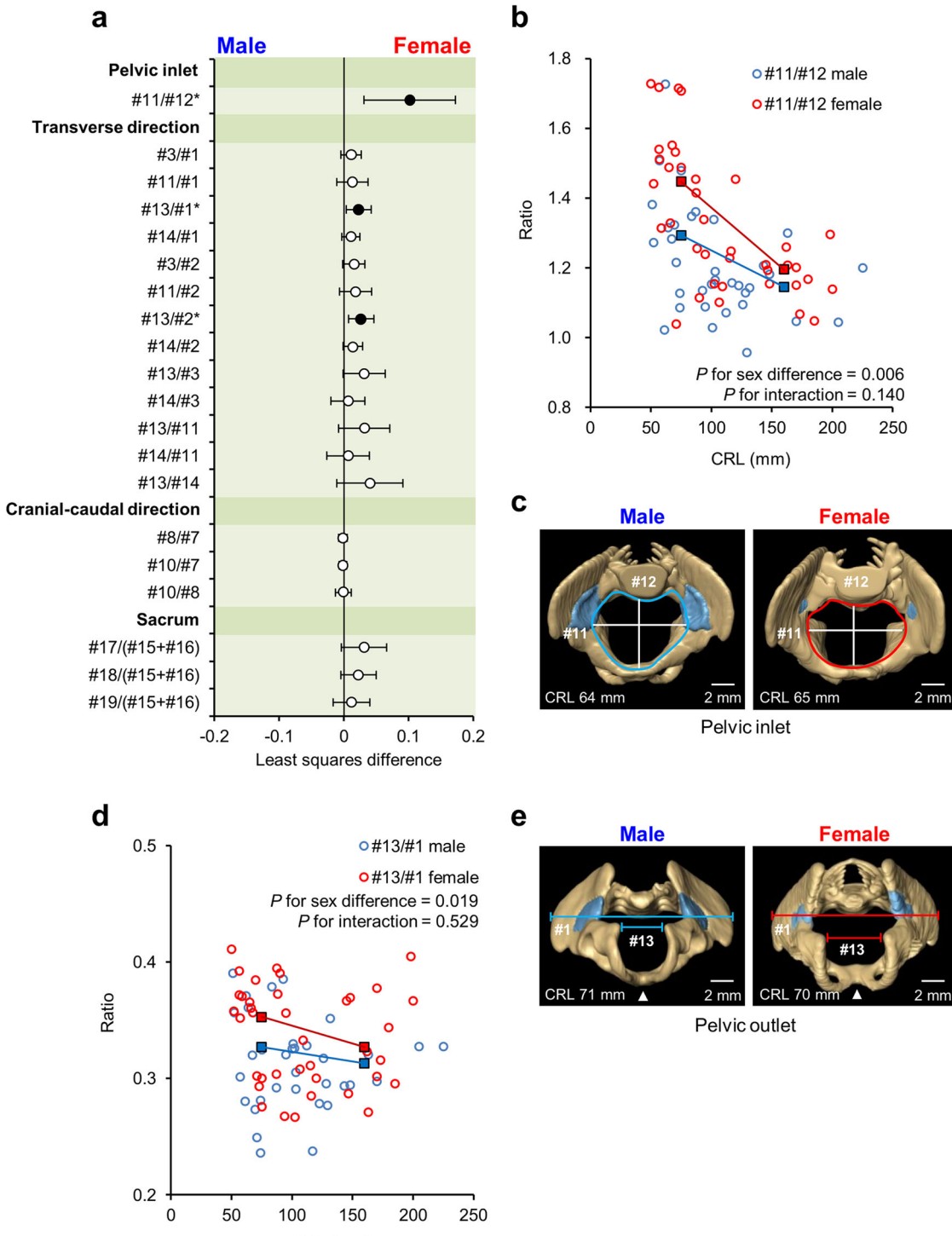

**Fig. 5 | Sexual dimorphism in pelvic ratios. a** Least squares mean and 95% confidence intervals for sex differences in the pelvimetry ratios. Positive least squares mean values indicate that females exhibit larger measurements. The asterisks and black circles indicate significant differences ($P < 0.05$). The dataset comprised 72 human fetus specimens. **b** Scatter plot of the pelvic inlet ratio (#11/#12). **c** Cranial view of the reconstructed pelvic inlet in males (CRL, 64 mm) and females (CRL, 65 mm). The light blue areas indicate the areas of primary ossification. Scale bar, 2 mm. **d** Scatter plot of the distance between the left and right ischial spines relative to the intercristal distance (#13/#1). **e** Caudal view of the reconstructed pelvic outlet in males (CRL, 71 mm) and females (CRL, 70 mm). The white arrowhead indicates the subpubic angle. Scale bar, 2 mm. Blue and red squares indicate the estimated means for males and females in each CRL subgroup, respectively. CRL crown-rump length.

the development and physiological processes of many structures other than the pelvis (cartilage, muscles, and ligamentous tissues)[33,40]; therefore, physiological features other than those of the pelvis should also be considered to explain the partial sex differences observed in the present study.

Sex differences in the fetal pelvis are generally considered insufficient for reliably determining sex from the hip bones[17]. During the fetal development period examined in this study, size variations represented a large proportion of growth variation, potentially concealing the sex differences in

**Fig. 6 | Significant interactions between sex and CRL in the sacrum. a** Scatter plot of the transverse diameter of the fifth sacral vertebra (#19). The male and female regression lines are represented in blue and red, respectively. The dataset comprised 72 human fetus specimens. **b** Ventral view of the reconstructed sacrum in males (CRL, 170 mm) and females (CRL, 173 mm). Scale bar, 5 mm. **c** Scatter plot of the ratio of the first sacrum's transverse diameter to its longitudinal length (#17/ [#15 + #16]). Blue and red squares indicate the estimated means for males and females in each CRL subgroup, respectively. **d** Ventral view of the reconstructed sacrum in males (CRL, 57 mm) and females (CRL, 57 mm). Scale bar, 2 mm. CRL crown-rump length.

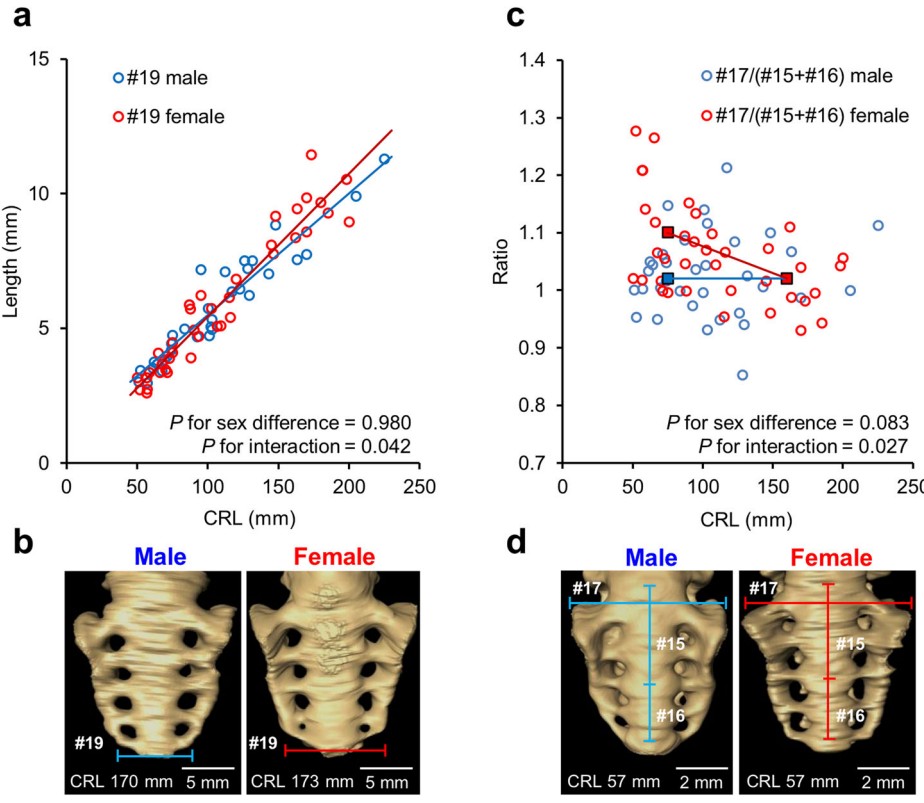

the pelvis. Thus, to accurately assess sexual dimorphism during periods of rapid growth, such as fetal development, both shape and size variations must be considered. Previous studies have analyzed sexual dimorphism in size-controlled fetal ilia[20,41]. Similarly, the present study evaluated pelvic sexual dimorphism using multiple regression analysis with CRL as a covariate to account for size variations. Moreover, this study revealed that ratios to evaluate the overall pelvic morphology could help detect sex differences that may be overlooked during individual measurements. These ratios may be beneficial during stages of fetal development when sex differences are unclear.

This study had some limitations. First, the possibility that some morphological and morphometric data were affected by fixation or the preparation process cannot be excluded. Second, since reliable sex determination by observation of the external genitalia was based on a CRL of 50 mm, the target for this study was set at a CRL ≥ 50 mm[28]. Determining the sex of a fetus with a CRL < 50 mm based on external genitalia observation is challenging. Therefore, if sex can be determined by histological or genetic analyses, it might be possible to study pelvic sexual dimorphism in fetuses with a CRL < 50 mm.

In conclusion, this study has reported the presence of sexual dimorphism at multiple sites in the human fetal pelvis at the onset of primary ossification earlier than previously reported. This study provides insights into the emergence of sexual dimorphism in the human pelvis, contributing to our understanding of human developmental phenomena and comparative anatomy.

## Methods
### Human fetus specimens
This study, which used human fetus specimens, was approved by the Kyoto University Faculty and Graduate School of Medicine Ethics Committee (Approval numbers: E986, R0316, and R2224). All ethical regulations relevant to human research participants were followed.

Approximately 44,000 human fetus specimens from the Kyoto Collection of Human Embryos are currently stored at the Congenital

Anomaly Research Center of Kyoto University[25] and Shimane University. Most were obtained from pregnancies terminated for socioeconomic reasons under the Maternity Protection Law of Japan. Parents provided verbal informed consent for their fetus specimens to be deposited in the collection, which was documented in the medical records; however, written consent was not obtained from all parents. The specimens were collected from 1963 to 1995 in accordance with the relevant regulations at each time point. Approximately 44,000 fetuses were stored, with 20% being well-preserved. Seventy-two human fetus specimens (34 male and 38 female) were included in this study (CRL 50–225 mm; age: approximately 9–23 weeks gestation)[24] (Supplementary Table 6). Fetal sex was determined by observing the external genitalia[28].

### Image acquisition
The MR images were acquired using 7-T MRI systems (BioSpec 70/20 USR; Bruker BioSpin MRI GmbH; Ettlingen, Germany, and Magnetom 7 T; Siemens Healthineers, Erlangen, Germany) and a 3-T MRI system (Magnetom Prisma; Siemens Healthineers, Erlangen, Germany).

The preclinical 7-T MRI system was employed to capture 3D images of fetal specimens ranging in CRLs from 50 to 88 mm. T1-weighted MR images were acquired using a fast low-angle shot pulse sequence[42]. The appropriate coil size and data acquisition settings were selected based on the size of the specimens[43]. Conversely, the clinical 7-T MRI system with a single-channel transmit and 28-channel receive knee coil (Quality Electrodynamics, OH, USA) or the 3-T MRI system was used for imaging fetus specimens with a CRL ≥ 88 mm[43,44]. During image acquisition, the fetus specimens were embedded in 1% agarose gel to fix their positions and avoid air bubble artifacts.

### Image data analysis
The MRI data from the selected fetus specimens were analyzed using serial 2D and reconstructed 3D images for accuracy. The 3D images were manually reconstructed using the Amira software version 5.5.0 (Visage Imaging GmbH, Berlin, Germany).

## Pelvimetry

Pelvic linear dimensions and angles were measured using the aforementioned 3D reconstructed images of the pelvis (Fig. 2, Table 1). The distances and angles evaluated were determined based on pelvimetry descriptions[25,45,46] and previous studies that investigated sexual dimorphism during the prenatal period and adulthood[23,47]. For values that could be measured symmetrically on both sides, the representative value was considered as the average of the left and right values. For pelvimetry of the greater pelvis, the intercristal (#1; distance between the furthest points of the two iliac crests) and interspinous (#2; distance between the anterior superior iliac spines) distances; the bilateral acetabula (#3), lateral conjugate (#4; distance between the anterior superior iliac spine and the posterior superior iliac spine), superior pubic ramus (#5; distance between the center of the acetabulum and the superomedial border of the pubic symphysis), pubic symphysis (#6; distance between the superomedial border and inferomedial border of the pubic symphysis), and os coxa (#7; distance from the most superior point on the iliac crest to the most inferior point on the ischial tuberosity) lengths; iliac blade height (#8; distance from the center of the acetabulum to the most superior point on the iliac crest); sciatic notch width (#9; distance from the posterior inferior iliac spine to the ischial spine), and ischial length (#10; distance from the center of the acetabulum to the most inferior point on the ischial tuberosity) were measured.

For pelvimetry of the lesser pelvis, the maximum transverse (#11) and the anteroposterior (#12; distance between the superior promontory of the sacral vertebra [S1] and the superomedial border of the pubic symphysis) diameters were measured to assess pelvic inlet formation, while the ischial spine diameter (#13, distance between the ischial spines) and the intertuberous distance (#14, distance between the ischial tuberosities) were measured to assess pelvic outlet formation.

Finally, the longitudinal length of the superior sacrum (#15, length between the sacral promontory and the midpoint of the third sacral vertebra), length of the inferior sacrum (#16, length between the midpoint of the third sacral vertebra and the inferior border of the fifth sacral vertebra), and transverse diameters of the first, third, and fifth sacral vertebrae (#17, #18, and #19) were measured to assess sacral growth.

For the greater pelvic angle, the iliac crest (#20) and subpubic angles (#21) were measured.

For the proportions of pelvic morphology, the pelvic inlet ratio of the (#11/#12), the transverse (#3/#1, #11/#1, #13/#1, #14/#1, #3/#2, #11/#2, #13/#2, #14/#2, #13/#3, #14/#3, #13/#11, #14/#11, #13/#14) and cranial-caudal (#8/#7, #10/#7, #10/#8) pelvic diameters, and the aspect length-to-width of the sacrum (#17/[#15 + #16], #18/[[#15 + #16], #19/[#15 + #16]] was calculated (Fig. 2, Table 1).

## Statistics and reproducibility

Statistical analyses were performed using the JMP software (JMP®, Version 16 Pro; SAS Institute Inc., Cary, NC, USA). A correlation coefficient was used to determine the correlation between each pelvimetric parameter and the CRL.

Multiple regression analysis was performed to examine the potential sexual dimorphism in pelvimetry, with each measurement or ratio set as the dependent variable. Since the linear dimensions were significantly correlated with CRL, a multiple regression model in which the independent variables were sex, CRL (as a continuous variable), and the interaction between sex and CRL was used. However, the correlation between the angle or ratio and the CRL was moderate or nonexistent. Therefore, a multiple regression model in which the independent variables were sex, subgroups based on the CRL (with a CRL of 100 mm as the boundary), and the interaction between sex and subgroups based on the CRL was used. The boundary of the CRL of 100 mm was established for two reasons: first, the trend of the distribution changed around a CRL of 100 mm for many angles and ratios (Supplementary Figs. 5–9, 11); and second, the CRL of 100 mm is the approximate size at the beginning of the second trimester[48]. Polynomial multiple regression analysis, including a squared CRL term, was also performed to verify the robustness of the results. Statistical significance

was defined as a two-tailed $P$ value of less than 0.05. Given the exploratory nature of this study, no corrections were made for multiple comparisons.

## Reporting summary

Further information on research design is available in the Nature Portfolio Reporting Summary linked to this article.

## Data availability

Supplementary Data 1 provides the source data behind the graphs presented in the manuscript. The corresponding author can provide additional data supporting this study's findings upon reasonable request.

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

## Acknowledgements

The authors thank Ms. Chigako Uwabe at the Congenital Anomaly Research Center and Dr. Akihiro Matsumoto at Shimane University for their technical assistance in handling human fetuses. The authors thank Dr. Tomohisa Okada of Kyoto University for the technical assistance in performing imaging using the clinical 7-T MRI scanner. This study was funded by the Japan Society for the Promotion of Science [Grant Nos. 20K22736, 21K07772, and 23K14976]. We would like to thank Editage (www.editage.com) for English language editing.

## Author contributions

Research design and study concept: T.K., J.M., T.T. Data analysis/interpretation: T.K., J.M., T.T. Acquisition of data: H.I., S.Y., H.O. Manuscript drafting: T.K. Manuscript review and editing: all authors.

## Competing interests

The authors declare no competing interests.
