## [Peer Review File · Communications Biology]

Reviewers' comments:

Reviewer #1 (Remarks to the Author):

Kanahashi and colleagues have investigated the appearance of sexually dimorphic features in the fetal pelvic bones of 72 specimens (34 males and 38 females) of the Kyoto collection (age range 9-23 weeks; at 9 weeks, sex determination based on the external genitals becomes possible). They used MRI to visualize 21 sites in the pelvis of these specimens. Furthermore, they calculated 20 ratios of these site measurements. The use of MRI assures the alignment of sections and, hence, the accuracy of the measurements. Although interesting and relevant, the topic seems challenging, because earlier investigations yielded conflicting outcomes on only a limited number of variables. Boucher (ref #19) found only sex dimorphism in the fetal subpubic arch and sciatic notch index, while others reported that the perinatal differences were only small compared to those seen around the age of highest fertility (20-40y; Huseynov et al, 2016). Haque et al (ref #23) observed that the subpubic angle was significantly different in 14-22-week-old fetuses, but not in older fetuses, whereas Hromada (ref #18) only observed such sex differences only after 30 weeks. Kanashi et al now report a shorter antero-posterior pelvic inlet and a larger subpubic angle in female fetuses, albeit that, in view of the number of specimens investigated, the significance level was modest ($P=0.036$ in both cases). The authors also demonstrated a larger distance between the ischial spines relative to the transverse diameters of the greater pelvis in female fetuses. Technically, the study was carried out meticulously. My reservations mainly deal with the presentation, in particular the extensive use of ratios and the definition of subgroups.

Comments on content:

- The anteroposterior diameter of the pelvic inlet (#12) was shorter in female than in male fetuses. That observation contrasts with the same measurement in adults, in whom the anteroposterior diameter of the pelvic inlet is reportedly shorter in males than in females (e.g. Correia et al, 2005; refs 30 & 31 also report: "In adults, the anteroposterior and transverse diameters of the inlet are significantly larger in females than in males". Please comment on this apparent difference.

- Somewhat unexpectedly, the relation of the observed sexual dimorphism with the "masculinization programming window" (e.g. Connan-Perrot et al, 2021 for review) is not discussed.

Comments on presentation:

1. The primary text shows mostly (least square) means rather than regression lines. For me, regression lines make more sense because they show the correlation between size (age) of fetuses and the parameter of interest, while sex differences would be visible as differences in the properties (position, inclination) of the regression lines. Age-dependent changes are of obvious importance in developmental studies. Least square means and 95% CI are valid numbers, but are also "dull" in that they suppress age effects and do not visually translate into biology. I provide a few examples which, in my view, show that these LSM do not help the presentation of the paper:

a. Ratios #11/#12 and #13/#1 in Figure 3C are good examples, as they show ratios which is analyzed after subclassification along the criteria shown in the panel "angles and ratios" of Figure 1, followed, according to the legend, by regression analysis. Just inspecting these panels, it is hard to imagine why regression analysis on least square means is more valid or relevant than regression analysis on the primary data.

b. Another example is the finding that "the transverse diameter of the middle pelvic cavity, where the ischial spine is located, relative to the transverse diameters of the greater pelvis (#13/#1, #13/#2) was significantly higher in females than in males" (lines 217-220). These ratios are just numbers which, even if statistically significant, seem to have little demonstrable biological significance.

I would, therefore, advocate to move the regression lines to the main text and the dull Tables to the supplementary data.

2. In describing sexual dimorphism in pelvic angles, the authors made subgroups. They do not explain why, and I cannot immediately see the logic. Given the gradual changes, I would have expected a non-linear regression analysis, with both curves showing when the described change occurs.

3. Ratios are modified primary data that are informative only if they make biological sense. This biological aspect is minimally discussed. The subgroups were mainly used to analyze ratios, which complicates the analysis, since the biological significance of the cut-off point (100 mm CRL, or approx. 13 weeks) is not clear.

a. An example is present in lines 184-5: "The transverse to anteroposterior diameter ratio at the pelvic inlet was significantly higher in females than in males". That may be true for fetuses <100 mm CRL, but looks problematic for the >100 mm CRL group. What is happening at that time point? On lines 199-200 it is stated that "significant sex differences in the anteroposterior diameters may have had a major influence on the significant differences in the ratio at the inlet". This example shows that ratios should be presented only if they offer a new biological insight.

b. Another example is the statement that "no sex differences were observed in the distance between the ischial spines, but a significant sex difference was observed in the relative distance between the ischial spines and the transverse diameter of the greater pelvis" (lines 224-227). What is the biological significance of this ratio over the primary data?

4. Many statistical analyses were carried out. The M & M section does not state whether the authors carried out a Bonferroni correction to compensate for their multiple comparisons.

5. The relevance of paragraph "lines 239-251" is not compelling.

6. Approximately 20% of the fetal specimens were well-preserved (lines 301-302): does this percentage refer to 44,000 or 72?

7. Figure 2, #1 and #2: "diameter" is probably "distance".

Reviewer #2 (Remarks to the Author):

The article entitled "Sexual dimorphism of the human fetal pelvis exists at the onset of primary ossification" is well written and extensively discussed. It can provide a different perspective on clinical anatomy and forensic medicine, and can be published without any corrections.

Referee: 1

We would like to extend our sincere gratitude to the Reviewer for their thoughtful comments and suggestions. The manuscript has significantly benefited from these insightful suggestions. Please find below our point-by-point responses to the Reviewer's comments.

Comments on content:

1) The anteroposterior diameter of the pelvic inlet (#12) was shorter in female than in male fetuses. That observation contrasts with the same measurement in adults, in whom the anteroposterior diameter of the pelvic inlet is reportedly shorter in males than in females (e.g. Correia et al, 2005; refs 30 & 31 also report: "In adults, the anteroposterior and transverse diameters of the inlet are significantly larger in females than in males". Please comment on this apparent difference

The anterior-posterior diameter of the pelvic inlet in males is longer than that in females during the early fetal period; however, the cause underlying this change remains unclear. We conducted a literature search to retrieve articles discussing the differences in the pelvic inlet from the late fetal period to childhood in terms of sex that reported findings similar to those observed in our study; however, we were unable to retrieve any relevant references.

We would be grateful to refer to any citations suggested by the Reviewer. We have stated our considerations in the revised manuscript as shown below (lines 205–215).

[Line 205-215]

In the present study, the anteroposterior diameters of the inlet were significantly larger in males than in females; however, the reason for this significance is unclear. No significant differences were observed in the transverse diameters of the inlet between the sexes. Thus, significant sex differences in the anteroposterior diameters may have had a major influence on the significant differences in the ratio at the inlet. The dilatation of the pelvic canal in adult females may be developmentally induced by the volume and location of the reproductive pelvic organs (vagina, ovaries, and uterus)³³

.

Therefore, it is necessary to investigate whether the location and size of the reproductive organs, rectum, bladder, and other pelvic organs affect sex differences in anteroposterior diameters in the fetal period.

2) - Somewhat unexpectedly, the relation of the observed sexual dimorphism with the “masculinization programming window” (e.g. Connan-Perrot et al, 2021 for review) is not discussed.

Thank you for your valuable comment. Following your comment, we have revised the Discussion section as shown below.

[Line 263–275]

Animal model developmental studies have revealed that pelvic sexual dimorphism is primarily determined by the spatial distribution of estrogen, androgen, and relaxin hormone receptors and hormone-induced bone remodeling^{36,37}

. Sertoli and Leydig

cells differentiate and secrete steroid hormones from the 8th and 9th weeks of gestation in humans^{38,39}. In addition, the masculinization programming window in humans lasts from the 8th to 14th weeks of gestation³⁹. Since this study was conducted on fetuses over a gestational age of 9 weeks, sex hormones may have influenced dimorphic growth and pelvic remodeling. Steroid hormones are involved in the development and physiological processes of many structures other than the pelvis (cartilage, muscles, and ligamentous tissues)^{33,40}; therefore, physiological features other than those of the pelvis should also be considered to explain the partial sex differences observed in the present study.

Comments on presentation:

1. The primary text shows mostly (least square) means rather than regression lines.

For me, regression lines make more sense because they show the correlation between size (age) of fetuses and the parameter of interest, while sex differences would be visible as differences in the properties (position, inclination) of the regression lines. Age-dependent changes are of obvious importance in developmental studies. Least square means and 95% CI are valid numbers, but

are also “dull” in that they suppress age effects and do not visually translate into biology. I provide a few examples which, in my view, show that these LSM do not help the presentation of the paper:

a. Ratios #11/#12 and #13/#1 in Figure 3C are good examples, as they show ratios which is analyzed after subclassification along the criteria shown in the panel “angles and ratios” of Figure 1, followed, according to the legend, by regression analysis. Just inspecting these panels, it is hard to imagine why regression analysis on least square means is more valid or relevant than regression analysis on the primary data.

b. Another example is the finding that “the transverse diameter of the middle pelvic cavity, where the ischial spine is located, relative to the transverse diameters of the greater pelvis (#13/#1, #13/#2) was significantly higher in females than in males” (lines 217-220). These ratios are just numbers which, even if statistically significant, seem to have little demonstrable biological significance.

I would, therefore, advocate to move the regression lines to the main text and the dull Tables to the supplementary data.

Thank you for your insightful suggestion. In accordance with your suggestion, we have summarized the scatter plots and regression lines for all measurements in the figures cited in the main text. However, we have presented these figures as data only the Editors and Reviewers can see (File name: 'Figure to be presented only to reviewers and editors') as we believe that these figures may not adequately convey our assertions to the readers. Please note that we have supplied this file only for your reference; we do not wish to publish these figures. We have fully considered the points raised and modified the tables and figures to enhance their readability.

The scatter plots and regression lines in the figure supplied originally were small and difficult to read. Therefore, we have divided the linear dimension, angle, and ratio results into three figures and enlarged each panel. Figures 3, 4, 5, and 6 present the linear dimensions, angle, ratio, and interaction results, respectively. The regression lines for all measurements have been supplied as Supplementary Figures. The tables presenting the

results of the statistical analysis have been supplied as Supplementary Tables. The description of the Results section has also been changed to reflect these changes in the figures. Furthermore, a subsection on sacral interactions has been added.

2. In describing sexual dimorphism in pelvic angles, the authors made subgroups. They do not explain why, and I cannot immediately see the logic. Given the gradual changes, I would have expected a non-linear regression analysis, with both curves showing when the described change occurs.

Thank you for your comment. At the outset, we considered two options for angle analysis: fitting a curve using polynomial regression or creating a subgroup term of CRL. The method with the subgroup term was chosen because it can easily incorporate an interaction term of sex and CRL in the regression model and can present the results in the same way as in the analysis of linear dimensions. The reasons for using a CRL of 100 mm as the boundary have already been described in lines 398 to 401 of the Methods section.

However, we considered it natural that there would be criticism of splitting a continuous CRL at a certain threshold. Therefore, we followed your suggestion and conducted two types of multiple regression analysis for the subpubic angle (#21); both included CRL as a continuous variable and one of which fitted a curve with a squared CRL term (i.e., polynomial regression). Significant sex differences were found in both analyses. Thus, the sexual dimorphism observed in our study is not dependent on the regression analysis model that divides the subgroups; it is a robust result demonstrating the sexual dimorphism when the regression analysis model is changed. We present the results of the added multiple regression analysis for the subpubic angle (#21) in Supplementary Table 4 and the polynomial multiple regression analysis results in Supplementary Table 5. In addition, in the Results section, lines 147 through 152, we described the results as consistent with the multiple regression analysis using subgroup terms and the additional multiple regression analysis.

[Lines 147-152]

To confirm the reliability of the sex difference identified in the subpubic angle (#21),

we employed two different types of a multiple regression analysis. Both analyses incorporated the CRL as a continuous variable, with one model incorporating a curve by introducing a squared CRL term. This additional regression analysis yielded consistent results (Supplementary Tables 4 and 5).

3. Ratios are modified primary data that are informative only if they make biological sense. This biological aspect is minimally discussed. The subgroups were mainly used to analyze ratios, which complicates the analysis, since the biological significance of the cut-off point (100 mm CRL, or approx. 13 weeks) is not clear.

Thank you for your comment. Ratios are commonly used in the analysis of sexual dimorphism in the shape of the pelvis in humans and other animals (Moffett et al., 2013; Fischer et al., 2017; Lorenzon et al., 2021; Yilmaz et al., 2024). Therefore, pelvic sexual dimorphism was also analyzed using ratios in our study. The ratio eliminates size information and enables the comparison of the changes in the shape of the pelvis between males and females. In particular, information regarding the changes in shape may be masked by information regarding the increase in size in morphometric analysis of the developing fetal period, making it difficult to evaluate the changes in shape when only linear dimensions are used as an indicator. The analysis using pelvic ratios in this study is of biological importance because it captures the shape. It facilitates comparison of the changes in the shape of the pelvis between the sexes. We have added a statement regarding the evaluation performed using ratios in the Discussion section.

[Lines 284–288]

Moreover, this study revealed that ratios to evaluate the overall pelvic morphology could help detect sex differences that may be overlooked during individual measurements. These ratios may be beneficial during stages of fetal development when sex differences are unclear.

The reason for using subgroups to analyze ratios is the same as for angles. As with the subpubic angle (#21), we added multiple regression and polynomial multiple regression analyses for ratios that showed significant sex differences (#11/#12, #13/#1, #13/#2). In

both analyses, we consistently found significant sex differences. Those results are summarized in Supplementary Tables 4 and 5 and are presented in lines 164-165 of the Result section.

[Lines 164-165]

The additional regression analysis yielded consistent results (Supplementary Tables 4 and 5).

The justification for using CRL of 100 mm as boundaries is described in the Methods section (lines 398-401). In short, there are two reasons. First, the trend of the distribution changed around a CRL of 100 mm for many ratios (Supplementary Figures 6,7,8,9,11); and second, the CRL of 100 mm is the approximate size at the beginning of the second trimester (O'Rahilly and Müller. 2006).

[Lines 398–401]

The boundary of the CRL of 100 mm was established for two reasons: first, the trend of the distribution changed around a CRL of 100 mm for many angles and ratios (Supplementary Figures 5,6,7,8,9,11); and second, the CRL of 100 mm is the approximate size at the beginning of the second trimester⁴⁸

a. An example is present in lines 184-5: “The transverse to anteroposterior diameter ratio at the pelvic inlet was significantly higher in females than in males”. That may be true for fetuses <100 mm CRL, but looks problematic for the >100 mm CRL group. What is happening at that time point? On lines 199-200 it is stated that “significant sex differences in the anteroposterior diameters may have had a major influence on the significant differences in the ratio at the inlet”. This example shows that ratios should be presented only if they offer a new biological insight. Thank you for your comment. Regarding #11/#12, we have no information on what happened at a CRL of 100 mm. However, as the interaction of sex and CRL was insignificant ($P = 0.140$) for this ratio, we cannot conclude that the sex difference seen in the small subgroup (CRL <100 mm) disappeared in the large subgroup (CRL \geq 100 mm). Therefore, it is challenging to discuss this point.

b. Another example is the statement that “no sex differences were observed in the distance between the ischial spines, but a significant sex difference was observed in the relative distance between the ischial spines and the transverse diameter of the greater pelvis” (lines 224-227). What is the biological significance of this ratio over the primary data?

Thank you for your comment. The ratio eliminates size information and enables the comparison of the changes in the shape of the pelvis between males and females. We added a statement regarding the ratio evaluation in the Discussion section.

Sex differences in the ratio of the distance between the ischial spines to the transverse diameter of the greater pelvis are biologically important in that they indicate differences in the overall shape of the fetal pelvis from a CRL of ≥ 50 mm.

[Lines 284–288]

Moreover, this study revealed that ratios to evaluate the overall pelvic morphology could help detect sex differences that may be overlooked during individual measurements. These ratios may be beneficial during stages of fetal development when sex differences are unclear.

4. Many statistical analyses were carried out. The M & M section does not state whether the authors carried out a Bonferroni correction to compensate for their multiple comparisons.

Thank you for your comment. Bonferroni correction was not performed in the present study. We have clarified this point in the Methods section.

[Lines 404–405]

Given the exploratory nature of this study, no corrections were made for multiple comparisons.

5. The relevance of paragraph “lines 239-251” is not compelling.

Thank you for your comment. In accordance with your comment, we have added an interpretation of the interaction term in the Discussion section, as shown below.

[Lines 247–262]

No significant differences were found in the linear dimension of the transverse

diameter of the fifth sacral vertebra (#19) or in the ratio of the transverse diameter of the first sacral vertebra to the longitudinal diameter of the sacrum ($\#17/[\#15+\#16]$) between the sexes. However, significant interactions were observed between sex and CRL, indicating that the coefficients of CRL differ significantly between males and females. These significant interactions suggest differences in changes in sacral morphology between sexes during the prenatal period. Tague³⁵ reported that the width of adult female sacral vertebrae exceeded those of adult male sacral vertebrae. The ratios of the transverse diameters of the first, third, and fifth sacral vertebrae to the longitudinal diameter of the sacrum were higher in females than in males in the present study, albeit not statistically significant. In addition, compared with males, females had a significantly larger coefficient of the CRL in the linear dimension of the transverse diameter of the fifth sacral vertebra (#19). The sacral ratio index used in the present study may aid in understanding sacral sexual dimorphism during late fetal and postnatal development.

6. Approximately 20% of the fetal specimens were well-preserved (lines 301-302):
does this percentage refer to 44,000 or 72?

Thank you for your comment. This percentage refers to 44,000. We revised the statement in the Methods section, as shown below.

[Lines 316–317]

Approximately 44,000 fetuses were stored, with 20% being well-preserved.

7. Figure 2, #1 and #2: “diameter” is probably “distance”.

Thank you for your comment. Following your comment, we have replaced "diameter" with "distance" in the revised manuscript. The explanations for #1 and #2 in the text (line 352) have been revised accordingly

REVIEWERS' COMMENTS:

Reviewer #1 (Remarks to the Author):

Takakuwa and colleagues have addressed several of my comments, and have modified their text accordingly. I do regret that what I have called "dull" tables and subgroup analysis have prevailed in the revised version. That decision, however, is to be discussed between the authors and the editors.